# Long-Term—But Not Short-Term—Plasticity at the Mossy Fiber–CA3 Pyramidal Cell Synapse in Hippocampus Is Altered in M1/M3 Muscarinic Acetylcholine Receptor Double Knockout Mice

**DOI:** 10.3390/cells12141890

**Published:** 2023-07-19

**Authors:** Fang Zheng, Jürgen Wess, Christian Alzheimer

**Affiliations:** 1Institute of Physiology and Pathophysiology, Friedrich-Alexander-Universität Erlangen-Nürnberg, 91054 Erlangen, Germany; fang.zheng@fau.de; 2Molecular Signaling Section, Laboratory of Biological Chemistry, NIDDK, NIH, Bethesda, MD 20892, USA; jurgenw@niddk.nih.gov

**Keywords:** muscarinic acetylcholine receptors, hippocampal CA3 pyramidal cells, mossy fiber synapses, frequency facilitation, long-term depression, long-term potentiation

## Abstract

Muscarinic acetylcholine receptors are well-known for their crucial involvement in hippocampus-dependent learning and memory, but the exact roles of the various receptor subtypes (M1–M5) are still not fully understood. Here, we studied how M1 and M3 receptors affect plasticity at the mossy fiber (MF)–CA3 pyramidal cell synapse. In hippocampal slices from M1/M3 receptor double knockout (M1/M3-dKO) mice, the signature short-term plasticity of the MF–CA3 synapse was not significantly affected. However, the rather unique NMDA receptor-independent and presynaptic form of long-term potentiation (LTP) of this synapse was much larger in M1/M3-deficient slices compared to wild-type slices in both field potential and whole-cell recordings. Consistent with its presynaptic origin, induction of MF-LTP strongly enhanced the excitatory drive onto single CA3 pyramidal cells, with the effect being more pronounced in M1/M3-dKO cells. In an earlier study, we found that the deletion of M2 receptors in mice disinhibits MF-LTP in a similar fashion, suggesting that endogenous acetylcholine employs both M1/M3 and M2 receptors to constrain MF-LTP. Importantly, such synergism was not observed for MF long-term depression (LTD). Low-frequency stimulation, which reliably induced LTD of MF synapses in control slices, failed to do so in M1/M3-dKO slices and gave rise to LTP instead. In striking contrast, loss of M2 receptors augmented LTD when compared to control slices. Taken together, our data demonstrate convergence of M1/M3 and M2 receptors on MF-LTP, but functional divergence on MF-LTD, with the net effect resulting in a well-balanced bidirectional plasticity of the MF–CA3 pyramidal cell synapse.

## 1. Introduction

Muscarinic acetylcholine receptors (mAChRs) have been long recognized as essential players in cognitive functioning [1,2,3,4,5], yet the particular roles of the five mAChR subtypes (M1–M5) are still not sufficiently resolved. Based on their downstream signaling pathways, mAChRs fall into two groups. The unevenly numbered receptors (M1, M3, M5) couple to G_q/11_ (M1-type receptors), whereas the evenly numbered receptors (M2, M4) signal via G_i/o_ (M2-type receptors) [3,6,7]. In the absence of highly subtype-specific pharmacological tools, the advent of knockout (KO) mice lacking one or two mAChRs substantially advanced the field, enabling a more detailed analysis of muscarinic effects on cognitive performance [8,9,10]. For example, M2-KO, but not M4-KO, mice exhibit deficits in hippocampus-dependent learning tasks [11,12,13,14]. Likewise, global and hippocampal-specific deletion of M3 receptors impairs learning and memory [15,16], whereas M1-KO mice show only selective deficits in tasks involving hippocampal–cortical interplay [17,18].

As long-term synaptic plasticity is widely accepted as a neurobiological substrate of learning and memory, an obvious question is whether the cognitive deficits of mAChR-KO mice can be attributed to impaired plasticity at the synaptic level. Indeed, for M2-KO mice, we found a significant decline in NMDA receptor-dependent long-term potentiation (LTP) in the hippocampal slices, both at the Schaffer collateral (SC)–CA1 synapse and the associational/commissural fiber (A/C)–CA3 synapse [12,19]. Less clear effects on LTP at the SC–CA1 synapse were observed in the hippocampal slice from M1- and M3-KO mice. Lack of M3 receptors did not alter LTP, whereas lack of M1 receptors led to either normal or reduced LTP, depending on the induction protocol [18,20,21]. Interestingly, we reported earlier that a lack of M2 receptors diminished LTP at the A/C–CA3 pyramidal cells (v.s.) and enhanced NMDA receptor-independent LTP at the mossy fiber (MF)–CA3 pyramidal cell synapse [19], suggesting that M2 receptors can regulate the strength of the two main projections onto CA3 pyramidal cells in an opposite, input-specific fashion.

Like other excitatory synapses, MF–CA3 synapses undergo long-term depression (LTD) following prolonged low-frequency stimulation (LFS) [22]. LTD is the counterpart of LTP, and its importance for cognitive processes is increasingly appreciated [23,24]. Like LTP, LTD at MF–CA3 synapses is predominantly NMDA receptor-independent and presynaptic in origin [22,25,26]. Since CA3 and dentate gyrus (DG) express appreciable levels of M1 and M3 receptors, but not M5 receptors [1,27,28], we took advantage of M1/M3 double KO (M1/M3-dKO) mice to explore how M1-type receptors shape lasting upward and downward changes in synaptic strength at the MF–CA3 synapse. Our data demonstrate that, in hippocampal slices from M1/M3-dKO mice, LTP is enhanced at the expense of LTD, which is abrogated. By contrast, elimination of M2 receptors augmented both LTP and LTD. Taken together, our data demonstrate that M1- and M2-type receptors regulate LTP and LTD at the MF–CA3 synapse in a synergistic and antagonistic fashion, respectively.

## 2. Materials and Methods

M1/M3-dKO mice (genetic background 129J1 × CF1) were generated as previously described [29]. In some experiments, homozygous M2 single KO (M2-KO) mice [30] were used for comparison. For each knockout strain, age-matched wild-type (wt) mice of the matching genetic background were used in parallel as controls. Mice were housed under standard conditions. All procedures were conducted in accordance with the Animal Protection Law of Germany and the European Communities Council Directive of November 1986/86/609/EEC), and with approval of local Franconian government.

Transverse hippocampal slices (350 μm thick) were prepared from adult male or female mice (3–7 month-old, anesthetized with sevoflurane) and maintained as described previously [19,31]. The slices were then kept in modified artificial cerebrospinal fluid (aCSF) containing (in mM) 125 NaCl, 3 KCl, 1 CaCl_2_, 3 MgCl_2_, 1.25 NaH_2_PO_4_, 25 NaHCO_3_=, and 10 d-glucose at room temperature for at least 2 h before being used. Individual slices were transferred to a submerged chamber perfused with normal aCSF with 1.5 mM MgCl_2_ and 2.5 mM CaCl_2_ at 31 ± 1 °C, unless otherwise stated. All solutions were constantly gassed with 95% O_2_ − 5% CO_2_. Signals were filtered at 2 kHz and sampled at 20 kHz using a Multiclamp 700B amplifier together with Digidata 1440A interface and pClamp10.6 software (Molecular Devices, Sunnyvale, CA, USA). MiniDigi 1A and AxoScope 10 were used for low-resolution scope recording, sampled at 1 kHz. Drugs and chemicals were obtained from Tocris Bioscience (Bio-techne GmbH, Wiesbaden, Germany) and Sigma-Aldrich Chemie GmbH (Steinheim, Germany).

Whole-cell recordings of visualized CA3 pyramidal cells in dorsal hippocampal slices were performed in voltage-clamp mode with patch pipettes filled with (in mM) 135 K-gluconate, 5 HEPES, 3 MgCl_2_, 5 EGTA, 2 Na_2_ATP, 0.3 Na_3_GTP, and 4 NaCl (pH 7.3, adjusted by 1 mM KOH). Cells were held at −70 mV and all potentials were corrected for liquid junction potential (15.5 mV). Series resistance in whole-cell configuration was 5–20 MΩ and compensated by 60–80%. To monitor the excitatory synaptic drive onto CA3 pyramidal cells, spontaneously occurring excitatory postsynaptic currents (spEPSCs) were collected in the presence of the GABA_A_-receptor antagonist, picrotoxin (100 μM). In some cases, tetrodotoxin (TTX, 1 μM) was introduced to the perfusing solution to block action potential discharge, yielding miniature EPSCs (mEPSCs). To elevate the level of ambient acetylcholine in the slice tissue, acetylcholinesterase activity was inhibited by eserine (10 μM).

Constant-current pulses (width 0.1 ms) were delivered to a bipolar tungsten electrode located in the hilus to activate mossy fiber (MF) projection. The evoked MF EPSCs were monitored at 0.1 Hz. Stimuli were carefully adjusted at low intensities to minimize polysynaptic and/or A/C pathway activation of CA3 pyramidal cells. MF responses were characterized by their prominent feature of strong facilitation during short trains of repetitive stimulation [26]. LTP of MF–CA3 synapses was induced by high-frequency stimulation (HFS) at 100 Hz for 1 s, and repeated 3 times at an interval of 10 s. Long-term depression (LTD) of MF EPSCs was induced by low-frequency stimulation (LFS) at 1 Hz for 15 min. The stimulation intensity during LTP/LTD induction protocol was kept consistent with that of individual baseline. As long-term plasticity of A/C synapses is NMDA receptor-dependent, the NMDA receptor antagonist, d-2-amino-5-phosphonopentanoic acid (d-AP5, 50 μM), was present in all experiments on MF synaptic plasticity to prevent contamination from A/C responses. Peak amplitude of evoked EPSC was measured, and a threshold of 5 pA was set to define the events as failure or response. Given the highly dynamic amplitudes of evoked MF EPSCs [26,32,33], the magnitude of LTP/LTD was expressed as changes in: (i) failure rate, calculated by counting failures among the total events during baseline (pre-tetanus) or post-tetanus (1–20 min), and (ii) averaged peak amplitude of evoked EPSCs (without failure) before and 16–20 min after tetanus. Data were included only when the peak amplitude of evoked EPSCs was reduced >90% by the group II metabotropic glutamate receptor agonist DCG IV (2.5 μM) at the end of the experiment.

MF-LTP experiments were also performed using extracellular recordings in CA3 stratum lucidum, with aCSF containing high divalent ion concentrations (4 mM CaCl_2_ and 4 mM MgCl_2_) to reduce polysynaptic recruitment contamination [32,33,34]. The recording pipette for field postsynaptic potentials (fPSPs) was filled with modified aCSF, in which NaHCO_3_ was replaced by HEPES to avoid pH change. LTP of CA3 MF fPSPs was induced by tetanic stimulation at 25 Hz for 5 s in the presence of d-AP5 (50 μM) [31].

Data analysis was performed offline with Clampfit 10.6 (Molecular Devices, CA, USA). Peak amplitudes of evoked MF responses were measured and averaged over 30 s (for fPSPs) or 60 s (for EPSCs). Spontaneous events were detected using an automated event detection algorithm with an amplitude threshold set as 4* *σ*_noise_. In addition to the frequency of synaptic inputs, the amplitude and the kinetics of sp/mEPSCs were measured from averaged events, which were selected only if no other event occurred during rise and decay. Rise time was measured from 10% to 90% of the peak response. The decay of averaged currents was fitted with single exponential functions using the Levenberg–Marquardt nonlinear least-squares algorithm. Tau reflects the time required for spontaneous events to decay to 37% of its peak value.

Data were expressed as mean ± SEM. OriginPro 2018 G (OriginLab Corporation, Northampton, MA, USA) was used for statistics and figures. The Shapiro–Wilk test was used to assess the normality of data distribution, and the null hypothesis was accepted when *p*-value was larger than 0.05. Statistical comparisons were performed using unpaired or paired Student’s *t*-test and one-way or two-way analysis of variance (ANOVA), followed by Tukey’s post-hoc test, as appropriate. Significance was assumed for *p* < 0.05.

## 3. Results

### 3.1. M1/M3 Receptor Double KO Reduces Excitatory Synaptic Drive onto CA3 Pyramidal Cells

Firstly, we examined whether the genetic disruption of M1/M3 receptors affects basal excitatory neurotransmission in the CA3 region using whole-cell recordings of pharmacologically isolated EPSCs from CA3 pyramidal cells that were voltage-clamped at −70 mV. As illustrated in Figure 1A, spontaneously occurring EPSCs (spEPSCs) in control slices exhibited a frequency of 4.49 ± 0.53 Hz (*n* = 24 from 10 wt mice; Figure 1B), with an average peak amplitude of 44.49 ± 3.38 pA (Figure 1C). M1/M3-dKO led to a significant reduction in both frequency (*n* = 34 from 13 mice, 3.30 ± 0.31 Hz; *p* = 0.042) and peak amplitude (32.51 ± 1.87 pA; *p* = 0.004) of spEPSCs, whereas spEPSC kinetics remained unchanged (Figure 1A–C). Thus, loss of M1/M3 receptors brought about a strong attenuation of the overall excitatory synaptic drive onto CA3 pyramidal cells. Notably, the remarkable change in synaptic input in our mutant preparations was not accompanied by significant alterations in intrinsic electrophysiological properties of the CA3 pyramidal cells (wt, *n* = 24; M1/M3-dKO, *n* = 34), such as input resistance (wt, 256.46 ± 21.24 MΩ vs. M1/M3-dKO, 236.47 ± 17.62 MΩ, *p* = 0.471) and membrane capacitance (wt, 109.04 ± 5.36 pF vs. M1/M3-dKO, 97.62 ± 3.59 pF, *p* = 0.071).

We next used TTX (1 μM) to silence network activity and abrogate firing-driven glutamate release. Under this condition, we observed a pronounced decrease in the frequency of the remaining miniature EPSCs (mEPSCs) in wt CA3 pyramidal cells compared to the frequency of spEPSCs before TTX was added to the bathing solution (*n* = 5, from 4.89 ± 1.17 Hz spEPSC to 2.11 ± 0.53 Hz mEPSCs; paired *t*-test, *p* = 0.016). To elucidate the effect of ambient acetylcholine on mEPSCs frequency and the role of M1/M3 receptors therein, we performed recordings with the acetylcholinesterase inhibitor, eserine (10 μM), and the M2-type-preferring antagonist, gallamine (20 μM) [35,36], in the bath, in addition to TTX and GABA_A_ receptor antagonist, picrotoxin. We also included the GABA_B_ receptor antagonist CGP 55845 (1 μM) to rule out putative indirect effects of GABA_B_ receptors at the MF–CA3 pyramidal cell synapse [37]. The application of eserine for 1–3 min enhanced both the mEPSC frequency (*n* = 6, from 2.75 ± 0.50 Hz to 4.66 ± 1.01 Hz, paired *t*-test, *p* = 0.020; i.e., 163.70 ± 10.79% of control) and peak amplitude (from 43.47 ± 4.73 pA to 48.32 ± 3.68 pA, paired *t*-test, *p* = 0.049) (Figure 1D–F), without changes in mEPSC kinetics (10–90% rise time: 0.86 ± 0.04 ms vs. 0.88 ± 0.04, paired *t*-test, *p* = 0.540; decay tau: 5.02 ± 0.39 ms vs. 5.32 ± 0.24 ms, paired *t*-test, *p* = 0.139). As illustrated in Figure 1D, the eserine-induced enhancement of mEPSCs was reversible upon wash-out (2.60 ± 0.58 Hz, *p* = 0.507 vs. values before eserine). To examine whether the facilitation of synaptic transmission following the eserine-induced elevation of ambient acetylcholine levels is mediated solely by M1-type receptors, we further added the nonselective nicotinic AChR (nAChR) antagonist, mecamylamine (10 μM) [38], to the above cocktail of blockers. With nAChRs suppressed, eserine still caused a significant enhancement of mEPSC frequency (*n* = 7; from 2.80 ± 0.57 Hz to 3.48 ± 0.74 Hz, paired *t*-test, *p* = 0.009) and peak amplitude (from 38.86 ± 2.22 pA to 43.56 ± 2.65 pA, paired *t*-test, *p* = 0.029) (Figure 1E,F), without a change in mEPSC kinetics. However, since the relative increase in mEPSC frequency in the presence of mecamylamine amounted to only 123.51 ± 3.00% of control, which was significantly lower than in the absence of this inhibitor (*p* = 0.003), M1-type receptors and nicotinic receptors appear to jointly promote firing-independent vesicular glutamate release.

### 3.2. M1/M3-dKO Facilitates LTP of Mossy Fiber–CA3 Synapses

Whereas the above recordings provided new information regarding the overall impact of M1-type receptors on the spontaneous excitatory drive experienced by CA3 pyramidal cells, they did not differentiate the synaptic events with respect to their origin, be it mossy fibers, A/C fibers, or the perforant path. To focus on the MF input and examine how its rather unique plastic changes are modulated by mAChR subtypes, we placed a stimulation electrode into the MF pathway and monitored evoked EPSCs by means of whole-cell recordings from voltage-clamped CA3 pyramidal cells. The suppression of EPSCs after application of the metabotropic glutamate receptor agonist, DCG IV, at the end of the experiment served to confirm the selective activation of the MF pathway (Figure 2C). MF-evoked EPSCs are distinct from other excitatory synaptic responses in that they show a very strong facilitation upon short, repetitive stimulation at a relatively high frequency (e.g., 4 stimuli at 20 Hz; Figure 2A, inset)—a stimulus paradigm that partially mimics firing patterns of DG granule cells in vivo [39]. To quantify this signature facilitation between genotypes, we normalized the subsequent EPSC peak amplitudes to that of the first response in the train. As summarized in Figure 2A (wt, *n* = 20 from 8 mice; M1/M3-dKO, *n* = 20 from 8 mice), the strong facilitation during the four-stimuli trains was not affected by the absence or presence of M1/M3 receptors. Likewise, another prominent feature of MF synapses, namely frequency facilitation [26], which is defined as strong facilitation during sustained low-frequency stimulation (Figure 2B), remained unchanged in the absence of M1/M3 receptors (wt, *n* = 8 from 6 mice; M1/M3-dKO, *n* = 8 from 5 mice). The mean increase of evoked MF-EPSCs at 1 min of stimulation was 347.93 ± 51.33% in wt cells and 280.14 ± 25.45% in M1/M3-dKO cells (*p* = 0.256).

In striking contrast to short-term and frequency facilitation, which were M1/M3 receptor-independent, these receptors came into play when we examined long-term plasticity at the MF–CA3 synapse. For induction of LTP, we used a high-frequency stimulation protocol (HFS at 100 Hz for 1 s, repeated 3 times with 10 s intervals). Lack of M1/M3 receptors led to a much larger potentiation of the evoked responses after HFS compared to the relatively modest potentiation observed in wt hippocampi (Figure 2C–F). As illustrated in Figure 2C,D, the responses of MF-CA3 synapses to a given stimulus varied in size, with failure rates (ratio of non-responsive stimuli vs. total stimuli during baseline) depending on stimulation intensity in individual slices. HFS engendered a massive reduction in failure rate in both groups (wt, *n* = 7 from 5 mice, from 24.90 ± 6.55% to 4.36 ± 1.85% over 20 min after HFS, paired *t*-test, *p* = 0.020; M1/M3-dKO, *n* = 8 from 5 mice, from 19.37 ± 6.39% to 1.01 ± 0.73% over 20 min after HFS, paired *t*-test, *p* = 0.018) (Figure 2C–E). In control hippocampi, HFS enhanced the averaged amplitudes of evoked MF-EPSCs to 143.73 ± 5.91% (*n* = 7), which was measured over 16–20 min post-HFS (Figure 2F). By contrast, the mean potentiation of MF-EPSCs in M1/M3-dKO hippocampi at the same time period reached 287.03 ± 32.33% (*n* = 8), which was significantly different from wt hippocampi [two-way- ANOVA, factor genotype F(_1, 279_) = 5.770, *p* = 0.000; factor time for 1–20 min F(_19, 279_) = 2.221, *p* = 0.004; Figure 2F). Since we observed a similarly pronounced increase in LTP for the same synapse in M2-deficient hippocampi [19], both M1/M3 and M2 receptors appear to constrain MF LTP.

Consistent with the presynaptic site of MF–LTP [26], we observed a strong increase in spontaneous synaptic events between the three HFS stimulus trains and immediately after them (Figure 3A). Specifically, the frequency of spEPSCs in wt pyramidal cells (*n* = 7) increased to 242.21 ± 29.55% [Figure 3B; from 2.28 ± 0.46 Hz to 4.72 ± 0.96 Hz within 30 s after HFS, then decayed quickly to 3.11 ± 0.58 Hz in the next 30 s; one-way ANOVA followed by Tukey’s post-hoc test, F(_1, 7_) = 29.39, *p* = 0.001], accompanied by an increase in peak amplitude [Figure 3C; from 23.80 ± 2.27 pA to 31.01 ± 4.24 pA within 30 s after HFS, i.e., 128.65 ± 9.50% of control; one-way ANOVA, F(_1, 7_) = 75.57, *p* = 5.34e^−5^]. In M1/M3-deficient slices, the HFS-associated rise in the number of spontaneous events sky-rocketed to 1035.00 ± 215.30% of the control (1.10 ± 0.17 Hz, *n* = 8), with a concomitant enhancement of spEPSC amplitude (184.11 ± 15.75% of control value 15.18 ±1.38 pA) (Figure 3B,C). The massive impact of M1/M3-deficiency on the responsiveness of CA3 spEPSCs to MF tetanus was evident from the plots of the relative changes that occurred immediately after HFS (Figure 3B,C, right; two-way ANOVA followed by Tukey’s post-hoc test, with factor 1—genotype and factor 2—time), for both frequency [F(_1, 28_) = 21.22, *p* = 8.11e^−5^] and peak amplitude [F(_1, 28_) = 21.74, *p* = 6.98e^−4^].

Next, we asked how the synaptic effects of M1/M3 receptors observed in single, voltage-clamped CA3 pyramidal cells would influence the collective neuronal behavior in a network with intact GABAergic inhibition. To address this issue, we performed field potential recordings in CA3 stratum lucidum in the absence of any blockers. The electrical stimulation of MFs evoked a characteristic biphasic response, in which a fiber volley (FV, reflecting synchronized action potential firing in the MF pathway) was followed by a field postsynaptic potential (fPSP, Figure 4A, inset). Again, suppression of fPSP by the metabotropic glutamate receptor agonist DCG IV demonstrated selective activation of the MF pathway. As we have previously reported [31], tetanic stimulation of MFs at 25 Hz for 5 s produced robust LTP in control slices, with peak fPSP amplitudes at 26–30 min post-tetanus increasing to 144.22 ± 7.80% of the control (0.24 ± 0.02 mV, *n* = 9 from 6 wt mice; Figure 4A,B). In M1/M3-dKO slices, the same protocol enhanced fPSP amplitude to 190.18 ± 22.29% of the control (0.18 ± 0.02 mV, *n* = 6 from mutant mice; Figure 4B), which was significantly stronger than in the wt counterparts (*p* = 0.037). In line with the corresponding whole-cell recordings (Figure 2A,B), fPSP recordings did not reveal significant differences between the genotypes regarding quadruple-pulse facilitation and frequency facilitation (Figure 4C,D).

### 3.3. M1/M3-dKO turns LTD into LTP

Like many other glutamatergic synapses in the CNS, MF synapses onto CA3 pyramidal cells undergo LTD after prolonged low-frequency stimulation (LFS at 1 Hz for 15 min). To study and compare muscarinic modulation of MF-LTD vs. -LTP at the same cellular level, we went back to whole-cell recordings of pharmacologically isolated MF-evoked EPSCs. Since MF-LTD is presynaptic in origin, as is MF-LTP, the whole-cell configuration should not interfere with either form of long-term plasticity at this particular synapse, in particular since NMDA receptors were blocked again with d-APV (see Methods). A typical example of MF-LTD in wt hippocampus is illustrated in Figure 5A. When determined 16–20 min after LFS, the average EPSC amplitude was reduced to 64.46 ± 9.57% of the control in wt hippocampi (*n* = 7 slices from 5 mice; Figure 5D), which was accompanied by enhanced failure rates. (Figure 5C). Note that in LTD experiments, control stimulation intensity before LFS was adjusted to obtain a low failure rate (4.31 ± 1.66% in wt slices, *n* = 7). This enabled the appropriate capture of the higher failure rates after LFS-induced LTD, which were significantly increased to 16.44 ± 4.81% (paired *t*-test, *p* = 0.030; Figure 5C). In striking contrast, application of the very same stimulation protocol not only abrogated LTD in M1/M3-dKO hippocampi, but even induced LTP, with MF-EPSC amplitudes increasing to 120.74 ± 5.38% of the control value 16–20 min after LFS (*n* = 5 from 4 mice, *p* = 0.001 vs. wt; Figure 5B–D).

Given that LTP of the MF–CA3 synapse in the hippocampus of M2-KO mice was augmented in a fashion strongly resembling that reported here for M1/M3-dKO mice [19], we wondered whether such synergism between M1- and M2-type receptors also holds for MF-LTD. To our surprise, we obtained the opposite finding: the extent of LTD in M2-KO slices was significantly enhanced, amounting to 58.40 ± 6.99% reduction (*n* = 6 slices from 4 M2-KO mice), compared to 28.01 ± 6.98% in wt slices (*n* = 5 from 4 mice; *p* = 0.031) (Figure 5E,F). LFS-induced LTD was also accompanied by a higher failure rate in the mutant cells (wt, from 5.84 ± 1.69% to 13.44 ± 4.55%, paired *t*-test, *p* = 0.107; M2-KO, from 1.42 ± 0.90% to 21.02 ± 6.98%, paired *t*-test, *p* = 0.026). Thus, M1/M3 and M2 receptors exert opposite effects on LTD at the MF–CA3 pyramidal cell synapse.

## 4. Discussion

Muscarinic depression of LTP at the MF–CA3 pyramidal cell synapse was reported first by Williams and Johnston in 1988 (see also Maeda et al., 1993) [40,41]. Since then, the peculiar electrophysiological properties of this rather unique hippocampal synapse have been studied in great detail and have been shown to be related to learning and memory tasks involving pattern separation and/or completion [26]. In view of the wealth of data accumulated on the many uncommon features and functions of the MF–CA3 synapse within the hippocampal circuitry and in behavioral readouts, it is quite surprising that we still know relatively little about how and for what purposes this synapse is modulated by acetylcholine.

In view of the lack of muscarinic agonists and antagonists with pronounced subtype selectivity [36], the generation of subtype-specific mAChR-KO mice was a major step towards delineating the physiological functions of the M1–M5 receptors [8,42]. Nevertheless, one might ask whether the fact that these mice all have global mAChR-KOs, altering muscarinic effects in many tissues and organs including the brain, might compromise firm conclusions on the role of the respective mAChR subtype, as compared to a conditional KO. In the context of our study, we are aware of only one mAChR-KO with a remote impact on hippocampal neurophysiology, which was reported from M5-deficient mice [43]. Although expression of M5 receptors in CA3 and DG is negligible [27], CA3 pyramidal cells showed a significant reduction of spEPSC frequency in that study. This seemingly paradoxical finding has been attributed to the fact that M5-KO mice suffer from constitutive constriction of cerebral arteries, leading to neuronal atrophy and impaired synaptic connectivity in the hippocampus and elsewhere in the brain [43].

We report here the unexpected finding that in hippocampi of M1/M3-dKO mice, MF-LTP is significantly augmented when compared to wt hippocampi. This finding is corroborated by the fact that we observed anomalously enhanced MF-LTP in M1/M3-dKO hippocampi using two independent experimental settings with distinct induction protocols, namely (i) field potential recordings from hippocampal slices exhibiting intact network activity, and (ii) whole-cell voltage-clamp recordings from CA3 pyramidal cells, in which the GABA_A_ receptor blocker, picrotoxin, was routinely added to the bathing solution to obtain unambiguous measurements of EPSCs. These experiments strongly suggest that activation of M1-type mAChRs serves to curtail MF-LTP.

Although quite obvious from the experimental evidence, this conclusion seems counterintuitive for two reasons. Firstly, as noted already by Williams and Johnston in their 1988 paper [40], muscarinic depression of MF-LTP would not have been predicted on the basis of the widely documented essential role of the cholinergic system in facilitating cognitive functions, including hippocampus-dependent learning and memory. Common wisdom links a decline in LTP to impaired cognitive performance. This relationship indeed holds for M2-deficient mice, whose memory deficits were attributed to reduced plasticity at the Schaffer collateral–CA1 synapse [12]. The second reason, why the above conclusion is puzzling, is based on the observations that hippocampi from both M1/M3-deficient and M2-deficient mice exhibit a strikingly similar increase in MF-LTP, as demonstrated here and in an earlier study [19], respectively. How might signaling pathways as different as those of M2-type receptors, which couple to G_i/o_ proteins, and those of M1-type receptors, which couple to G_q/11_ proteins, functionally converge on inhibition of MF-LTP?

In the hippocampus, M1 and M3 receptors are mainly located postsynaptically [1], where they target various ion conductances to enhance cell excitability and promote firing. These mechanisms include the suppression of K^+^ currents, such as M-current (I_m_) and a slow Ca^2+^-activated K^+^ current (I_AHP_), and increase of depolarizing cation currents, such as the hyperpolarization-activated current (I_h_) and a Ca^2+^-dependent nonspecific cation conductance (I_cat_) [44,45,46]. Using mice lacking M1 receptors, Fisahn et al. demonstrated that M1 receptor activation depolarizes CA3 pyramidal cells by increasing I_h_ and I_cat_ [47]. Thus, reduced muscarinic excitation of presynaptic granule cells and CA3 neurons most likely accounts for the diminished spEPSC frequency that we measured in CA3 neurons from M1/M3-dKO hippocampi.

Do M1-type receptors also have a presynaptic site of action to regulate glutamate release directly? We addressed this issue by monitoring mEPSCs in the presence of TTX and pharmacological suppression of GABA_A_, GABA_B_, M2-type, and nicotinic receptors. When we enhanced the level of ambient acetylcholine with the acetylcholinesterase inhibitor eserine, we observed a significant increase in mEPSC frequency, which is most likely mediated by presynaptic M1 receptors. In support of this notion, M1 receptors have indeed been found to distribute along mossy fibers, albeit at lower densities compared to those in dendrites and spines [48]. Note that, although MF-LTP is presynaptic, M1 receptors do not necessarily have to reside on terminals to regulate the strength of synaptic potentiation. An attractive candidate pathway to account for the apparent disinhibition of MF-LTP in the absence of M1/M3 receptors involves retrograde endocannabinoid signaling. Activation of postsynaptic M1 and M3 receptors during strong synaptic use may trigger release of endocannabinoids from the postsynaptic site [49], which in turn bind to presynaptic CB1 receptors to suppress transmitter release [50].

Whereas it remains to be determined in future studies how postsynaptic and/or presynaptic M1/M3 receptor signaling contains MF-LTP, explaining how M2 receptor activation results in the same outcome seems more straightforward. The canonical pathway of MF-LTP comprises the following sequence [26]: Ca^2+^ influx through presynaptic voltage-dependent Ca^2+^ channels → activation of Ca^2+^-sensitive adenylyl cyclase 1 → elevation of cAMP levels → activation of PKA → persistent increase in transmitter release. As discussed in more detail previously, the presynaptic M2 heteroreceptors on MF terminals may interfere with LTP induction through inhibition of presynaptic Ca^2+^ channels and/or attenuation of adenylyl cyclase activity [19].

While M1 and M2 receptors seem to use different routes to curtail MF-LTP, our study also reveals some commonalities in the way they act. Firstly, with GABA_A_ receptors being routinely blocked in our whole-cell recordings, elimination of either mAChR subtype should not have disinhibited MF-LTP through a GABAergic mechanism, where activation of presynaptic GABA_A_ receptors facilitates MF–CA3 synaptic plasticity [51]. Secondly, in both field potential and whole-cell recordings, we employed robust stimulation protocols to induce presynaptic MF-LTP, instead of weak stimulation protocols, which induce an unorthodox postsynaptic and NMDA receptor-mediated form of MF-LTP [52,53]. Thus, M1 and M2 subtypes should both have a presynaptic site of action to regulate LTP (including retrograde signaling). Thirdly, both mAChR types not only inhibit MF-LTP, they also do not affect the unique hallmarks of MF short-term plasticity, namely quadruple-pulse facilitation and frequency facilitation.

Endowed with these latter features, MF synapses can act as a “conditional detonator” [54]. This particular property allows the MF synapse to assume a role as unsupervised “teacher” synapse, triggering plastic changes in the connectivity pattern of CA3 neurons. In the case of place cells, such formed ensembles of CA3 pyramidal cells are important for storage and recall of spatial information [54]. Put simply, muscarinic inhibition of MF-LTP might thus be envisioned as a means to preserve the integrity of the “conditional detonator”, which might unintentionally blow up when synaptic potentiation is not properly controlled.

Whereas MF-LTP is synergistically capped by activation of M1- and M2-type receptors, our study demonstrates that the two receptor types exert opposite effects on MF-LTD. In M1/M3-dKO, LTD was abrogated and LFS produced a small potentiation, whereas loss of M2 receptors augmented LTD (Figure 6). Interestingly, a very similar shift from LTD to LTP following LFS was observed in visual cortex slices from M1/M3-dKO mice [55]. Unlike MF synapses in the hippocampus, the excitatory synapses examined in the visual cortex preparation display postsynaptic, NMDA receptor-dependent long-term plasticity. It is remarkable that, although the sites and mechanisms of induction of LTP and LTD differ substantially between hippocampal MF synapses and the synapses in visual cortex, both synapses rely on M1/M3 receptor activation to prevent the paradoxical conversion of LTD to LTP following LFS.

Our data obtained with wt hippocampal preparations suggest that, under physiological conditions, the opposing forces that act on MF-LTD, namely M1/M3 receptor-mediated augmentation vs. M2 receptor-mediated inhibition, are matched to enable a degree of LTD that is capable of counterbalancing LTP. We indeed found that the long-term plasticity of the MF–CA3 pyramidal cell synapse extends almost equally in both directions, with a rather small bias in favor of LTP over LTD (Figure 6). For several reasons, it has been postulated that in a network where synapses undergo LTP, LTD is a necessary counterweight to enhance the overall performance in information processing, storage, and recall [56]. First and foremost, LTD counteracts the saturating effects that would ensue from potentiation alone. Furthermore, LTD facilitates the grouping of potentiated synapses that constitute a memory trace by suppressing synapses that do not participate in encoding this particular trace. Finally, LTD enables behavioral flexibility by weakening previously learned information that would interfere with the acquisition of new information in a changing environment.

This latter conclusion resulted from work with transgenic mice in which NMDA-dependent LTD of the Schaffer collateral–CA1 synapse was selectively disrupted [57]. A similar approach to decipher the functional role of MF-LTD has not been reported yet. However, valuable insights come from field potential recordings in freely behaving rats, demonstrating that LTD in the CA3 region encodes different aspects of a novel environment in an input-specific fashion: MF-LTD is associated with exploration of landmark objects, whereas exploration of discrete positional features of the environment facilitates A/C-LTD [58]. Whereas the full behavioral implications of MF-LTP and -LTD are only beginning to be understood, our study shows that muscarinic receptor activation confers a properly balanced bidirectional plasticity on the MF–CA3 pyramidal cell synapse, which should be important for optimal functionality and flexibility in learning and memory tasks.

The Bienenstock–Cooper–Munro (BCM) theory of synapse modification has become an influential concept to model and predict bidirectional synaptic plasticity at excitatory synapses [59]. Originally developed to account for synaptic modifications in the visual cortex of kittens following monocular deprivation [60], BCM theory was later extended to provide a formal description of the relationship between “classical” (NMDA receptor-mediated) LTD and LTP in the hippocampus and elsewhere. It remains to be examined, though, whether BCM theory is also applicable to the rather unique properties of MF-plasticity. Central to the BCM model is the biphasic plasticity induction function, φ, which calculates the likelihood that afferent synaptic activity induces either LTD or LTP, based on the correlated postsynaptic activity. Plotting the change in synaptic weight as a function of postsynaptic activity yields a characteristic curve, where LTD first waxes and wanes as postsynaptic activity gradually increases. Then, the curve crosses baseline and the synaptic weight change grows in the opposite direction (LTP)—until saturation. The intersection of the curve with the baseline, where the sign of synaptic plasticity reverses polarity, is termed the synaptic modification threshold, θ_m_. Importantly, BCM theory sets θ_m_ as a sliding threshold, thereby introducing a homeostatic mechanism, according to the following metaplastic rule: In a neuron with a prior history of strong firing, θ_m_ will be elevated, thereby impeding LTP and facilitating LTD; conversely, θ_m_ is decreased following a period of weak postsynaptic activity, now favoring LTP over LTD. Proposals on the mechanisms underlying the sliding θ_m_ all comprise postsynaptic effects, including changes in NMDA receptor subunit composition [61], in CaMKII levels [62], in Ca^2+^ release from intracellular stores [63], and in H-current activity [64].

To make our findings fit BCM theory, several points need to be considered. Although we did not vary MF stimulation systematically over a wide frequency range, it seems plausible to assume that MF-LTD and -LTP exhibit a relationship to presynaptic activity that can be described by a BCM-like curve. We further assume that the synaptic modification at the MF–CA3 pyramidal cell synapse displays a sliding threshold θ_m_ analogous to conventional synapses, but with a presynaptic mechanism—possibly involving the cAMP/PKA cascade [26,65]. Our data predict that activation of mAChRs is capable of moving θ_m_ towards higher values of presynaptic activity. This rightward shift of θ_m_ makes LTP more difficult, but, in contrast to the conventional BMC model, it also impedes LTD. Thus, application of a modified BMC theory to model muscarinic regulation of MF-plasticity seems feasible, provided that the above issues are addressed.

## 5. Conclusions

For a synapse such as the MF–CA3 pyramidal cell synapse, which operates as a conditional detonator, it is essential to prevent runaway potentiation when plasticity-inducing stimulation is repeated over time. We propose that the dense cholinergic innervation of the CA3 region is intimately involved in curtailing uncontrolled potentiation. In fact, acetylcholine secures this objective by recruiting mAChRs from both subtype families, underscoring its functional significance. In the framework of a modified BCM theory, the muscarinic effect on long-term MF-plasticity can be understood as a recalibration of the synaptic modification threshold, θ_m_, which is independent from metaplasticity. Notably, mAChR activation leaves intact the characteristic short-term plasticity of the MF–CA3 pyramidal cell synapse, as this feature appears indispensable to maintain its full operability within the hippocampal network. In summary, our study adds a missing piece to the greater picture of how the cholinergic system tunes the many neural properties of the hippocampus to promote cognitive functions.

## Figures and Tables

**Figure 1 cells-12-01890-f001:**
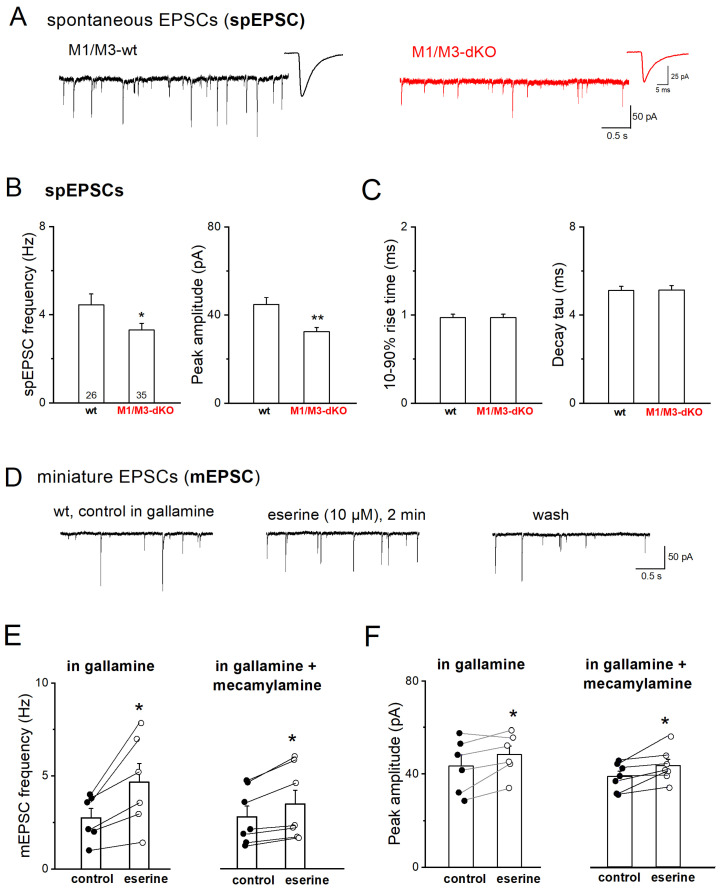
M1/M3-dKO reduces excitatory synaptic drive onto CA3 pyramidal cells. (**A**) Representative spEPSC recordings from wt neuron (black trace) and mutant neuron (red trace), respectively. Insets depict averaged synaptic events from respective cells. (**B**,**C**) Comparison of frequency and peak amplitude (**B**) and kinetics (**C**) of spEPSCs between the two genotypes. Numbers in columns indicate sample size. (**D**–**F**) Reversible increase in mEPSC frequency and amplitude during application of acetylcholinesterase inhibitor eserine (10 μM) in wt slices superfused with gallamine (20 μM) alone or in combination with mecamylamine (10 μM), in the presence of TTX (1 μM), picrotoxin (100 μM), and CGP 55845 (1 μM). * *p* < 0.05, ** *p* < 0.01.

**Figure 2 cells-12-01890-f002:**
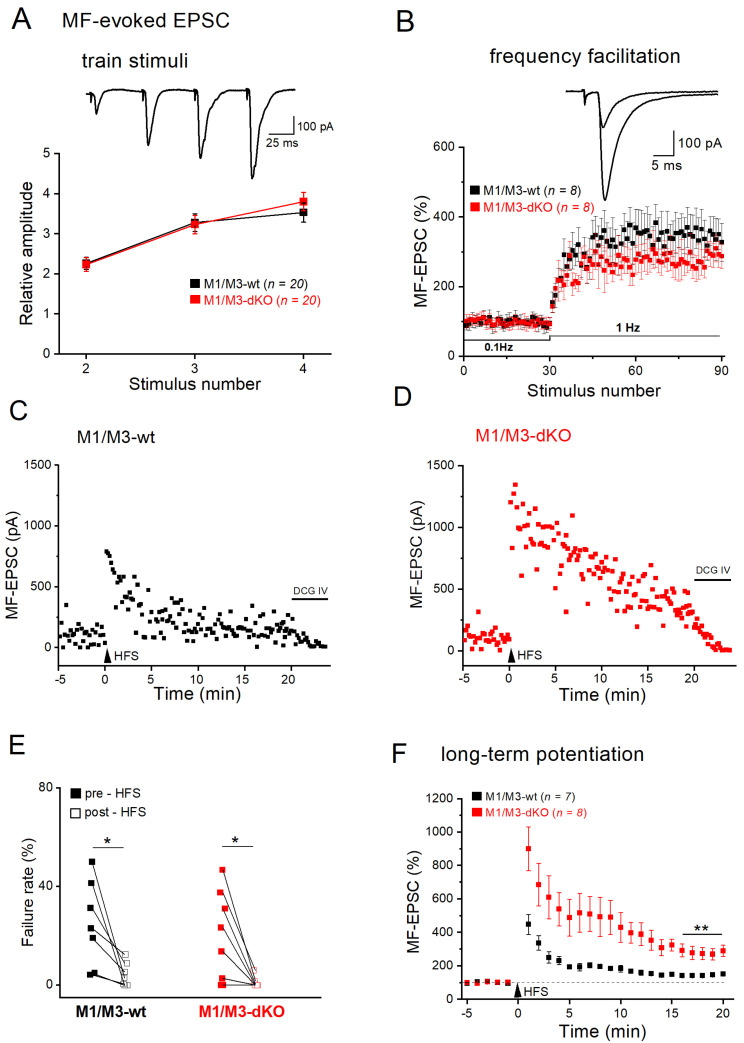
M1/M3-dKO does not alter STP, but strongly facilitates LTP of the MF–CA3 pyramidal cell synapse. (**A**,**B**) No significant change in short-term plasticity (STP) in M1/M3-dKO preparation. Top current trace in (**A**) from wt slice illustrates massive synaptic facilitation during quadruple-pulse stimulation with 55 μA. Diagram below summarizes EPSC amplitudes normalized to that of 1st EPSC. (**B**) Frequency facilitation after switching from 0.1 Hz to 1 Hz stimulation. EPSC amplitudes were normalized for comparison. Inset shows EPSC traces from a wt cell before and 1 min post 1 Hz stimulation. (**C**,**D**) Scatter plots of EPSC responses before and after HFS in wt pyramidal cell (**C**) and mutant pyramidal cell (**D**). (**E**) Decline in failure rates after HFS in both genotypes. (**F**) Comparison of HFS-induced LTP in wt cells (black symbols) and mutant cells (red symbols). * *p* < 0.05, ** *p* < 0.01.

**Figure 3 cells-12-01890-f003:**
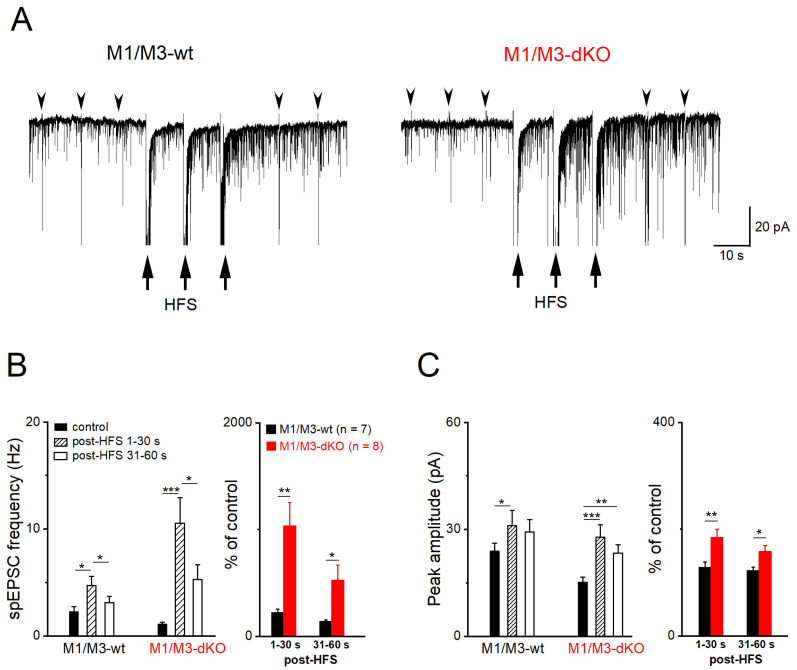
M1/M3-dKO augments spEPSC after HFS of MF pathway. (**A**) Current traces from wt and M1/M3-dKO pyramidal cells before, during, and after three HFS trains (arrows) to induce MF–LTP. Arrowheads indicate evoked EPSCs at 0.1 Hz. (**B**,**C**) HFS-induced changes in spEPSC frequency (**B**) and amplitude (**C**), quantified every 30 s. Statistical comparisons were conducted using one-way (left panels) or two-way (right panels) ANOVA, followed by Tukey’s post-hoc test at alpha = 0.05. * *p* < 0.05, ** *p* < 0.01, *** *p* < 0.001.

**Figure 4 cells-12-01890-f004:**
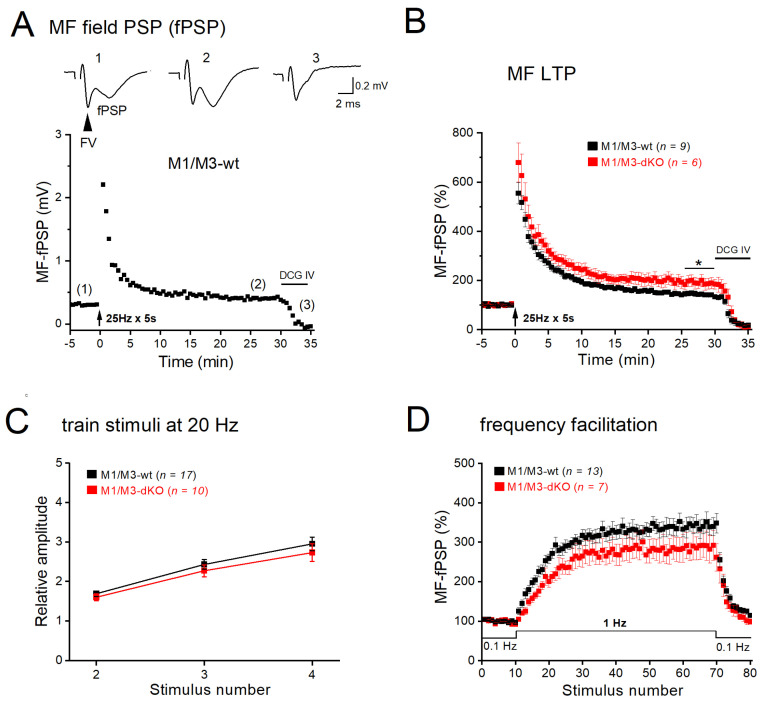
M1/M3-dKO promotes MF–LTP in extracellular recordings from CA3 stratum lucidum. (**A**) Trajectory of fPSPs before and after LTP-inducing stimulation (25 Hz for 5 s) in wt slice. Inset above depicts fPSP traces from like-numbered time points in trajectory. Stimulus artifacts were removed from fPSP traces. Arrowhead indicates fiber volley. (**B**) Comparison of normalized LTP trajectories between genotypes. Two-way ANOVA for 1–30 min post-tetanus: factor genotype, F(_1,840_) = 189.53, *p* = 4.94e^−39^. (**C**,**D**) M1/M3-dKO did not affect neither quadruple-pulse facilitation (**C**) nor frequency facilitation (**D**) of fPSPs. * *p* < 0.05.

**Figure 5 cells-12-01890-f005:**
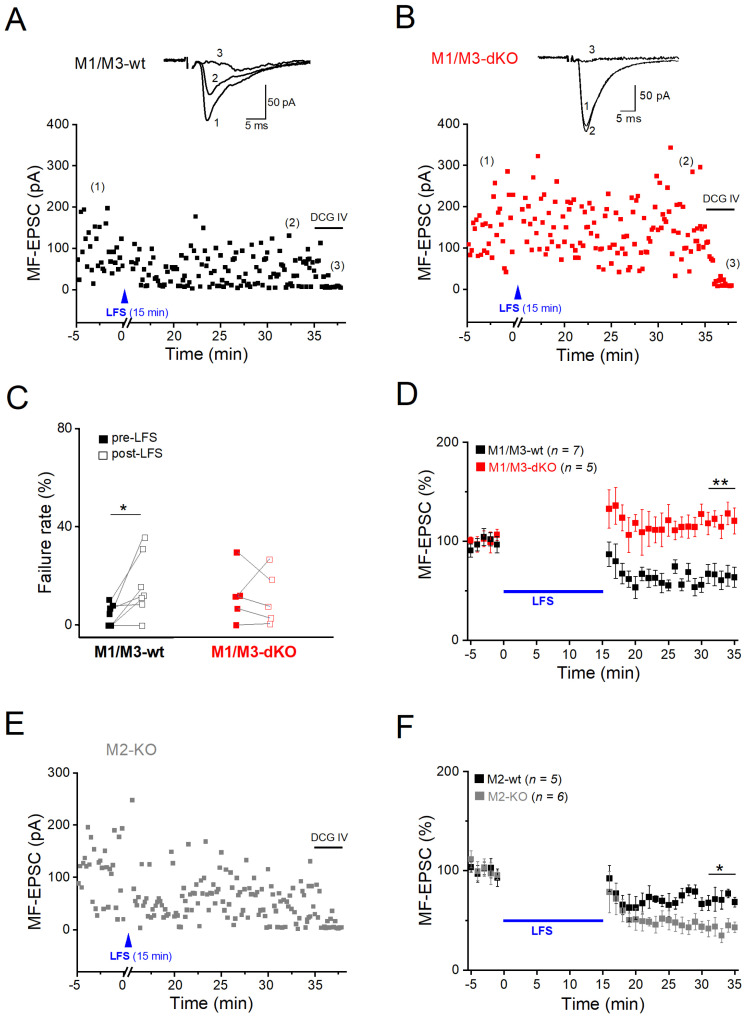
M1/M3-dKO converts LFS-induced LTD into LTP of the MF–CA3 pyramidal cell synapse. (**A**,**B**) Scatter plots of EPSCs before and after LFS in wt pyramidal cell (**A**) and M1/M3-dKO pyramidal cell (**B**). Insets above illustrate averaged EPSC traces from like-numbered time points. (**C**,**D**) Summary of changes in failure rates (**C**) and amplitudes (**D**) of EPSCs after LFS for either genotype. (**E**) Scatter plot of EPSCs before and after LFS in M2-KO preparation. (**F**) Comparison of LFS-induced MF-LTD between wt and M2-KO hippocampi, using normalized EPSC responses. Two-way ANOVA for 1–20 min post-LFS: factor genotype, F(_1,218_) = 134.99, *p* = 1.33e^−24^ (D); F(_1,199_) = 36.36, *p* = 7.83e^−9^ (**F**). * *p* < 0.05, ** *p* < 0.01.

**Figure 6 cells-12-01890-f006:**
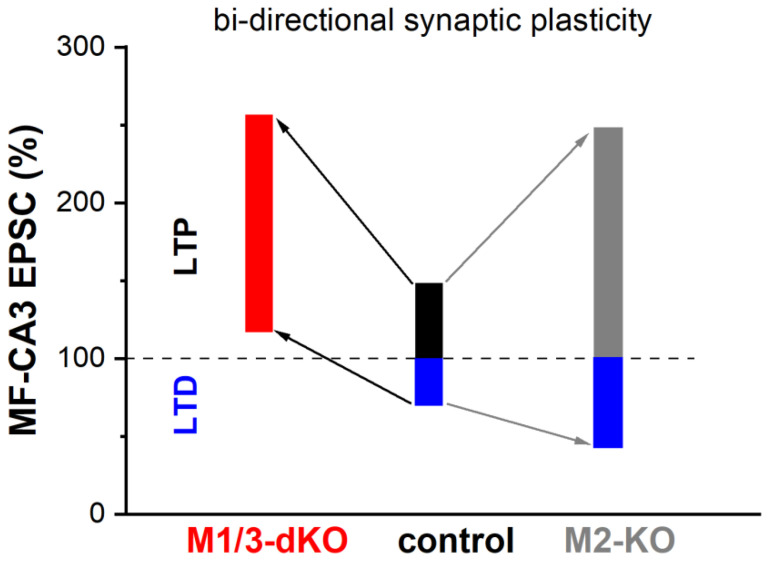
Synopsis of how mAChRs shape bidirectional plasticity at the MF–CA3 pyramidal cell synapse. See text for details.

## Data Availability

Data of this study are available from the corresponding author upon reasonable request.

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
