# Peer review of "Long-Term—But Not Short-Term—Plasticity at the Mossy Fiber–CA3 Pyramidal Cell Synapse in Hippocampus Is Altered in M1/M3 Muscarinic Acetylcholine Receptor Double Knockout Mice"

_cells, 2023, doi:10.3390/cells12141890_

Round 1

Reviewer 1 Report

Paper by Zheng and coworkers investigate the role of muscarinic M1/M3 receptors over plasticity at the mossy fiber pyramidal cell synapse.  Authors find an enhanced LTP and an abrogated LTD. Moreover elimination of M2 receptors augmented both LTP and LTD. Paper is well written and experimental approach sound appropriate. Some issues require attention 

Experiments reported in figure 3 seems quite a repetition of previous find and is not clear their purpose. Would have been more appropriate a Strontium based protocol to investigate the quantal massive release of neurotransmitter after synaptic stimulation

Would be interesting to pharmacological block M1/M3 receptors in WT mice in plasticity experiments. Have the authors tested this option? Conversely in spontaneous synaptic activity authors tested many antagonist but non one M1/M3 agonist in WT mice. Also here have the authors tested this option?

Short term plasticity could be improved by  paired pulse protocol with a variable inter-event intervals. Also input-output curve for sinaptic stimulation could be useful for testing the basic synaptic properties at WT and KO mice

Physiological interplay between LTD and LTP is not fully clear, discussion could benefit by a more extensive description

Could authors provide some reference about the specificity of gallamine binding for M2 receptors vs M1 

Many plots (fig 2B-D; fig 4B-D; fig 5A,B) repot labeling of X axis as “sample number”, is not clear what exactly represent, cause time o similar unit are more expected here. 

Reviewer 2 Report

Manuscript entitled “M1/M3 Muscarinic Acetylcholine Receptors Control Long-Term, but not Short-Term Plasticity at the Mossy Fiber - CA3 Pyramidal Cell Synapse in Mouse Hippocampus” by Zheng et al describes the role played by M1/M3 AChRs in hippocampal plasticity of mossy fiber synapses. The main goal set by the Authors of the manuscript is to investigate the function of M1/M3 muscarinic receptors in the regulation of synaptic transmission and its plasticity in CA3 region of the hippocampus. The Authors show that double knockout of M1/M3 receptors decreases frequency and amplitude of excitatory synaptic currents recorded in CA3 pyramidal cells. Whereas the increased concentration of extracellular ACh augments frequency and amplitude of mEPSC in CA3 PCs in a process that partially depends on M1/M3 receptors. These results suggest the modulatory impact of M1/M3 receptors on excitatory synaptic transmission in CA3. Next, Authors present results showing that double knockout of M1/M3 results in enhanced MF-CA3 LTP and a conversion of MF-CA3 LTD into LTP.

The manuscript is informative and clearly written. Presented discoveries would be suitable for researchers of synaptic transmission, neuromodulation and synaptic plasticity. I have two major comments and few minor ones.

Major comment:

1.      In M&M section Authors have described that ANOVA or Student's t-test was used for statistical comparisons. Please provide an information whether the normality of the distribution was assessed. Parametric tests cannot be used for data with non-normal distribution, thus aforementioned data normality assessment is crucial.

2.      The Authors performed an analysis of amplitude, frequency and kinetics of spontaneous EPSC recorded in CA3 PCs. It is unclear why similar analysis was not performed for miniature EPSC. In the opinion of the reviewer, the conclusions presented in the manuscript will be significantly strengthened after the presentation of data showing the impact of M1/M3 knockout on the amplitude, frequency and kinetics of mEPSCs. It will be also crucial to validate already presented data and check the impact of M1/M3 knockout on eserine effect.

I have couple minor comments as listed below:

1.      Page 3, line 5 - please provide detailed value of liquid junction potential used for correction of membrane potential

2.      Page 3, line 25 – Please specify which EPSC parameter were considered to calculate DCGIV- induced reduction to be at least 90%.

3.      Page 3 sentence “MF-LTP experiments were also performed using extracellular recording in CA3 stratum lucidum, with aCSF containing 4 mM CaCl2 and 4 mM MgCl2 to avoid multiple synaptic processes.” Bolded fragment is unclear. Please explain.

4.      Page 4, first new paragraph – presented data regarding input resistance and capacitance need to contain also statistical sample size

5.      Page 4, line 13 from the bottom “Application of eserine for 1-3 min reversibly enhanced both mEPSC frequency”. The manuscript does not contain data to justify the statement that serine reversibly change mEPSP parameters.

6.      Please provide information on how failure rate was calculated.

7.      In all experiments on synaptic plasticity Authors analysed changes in synaptic strength (efficacy) and changes in failure rates. Observed changes in synaptic efficacy may be a result of changed failure rate (as Authors show) and synaptic potency (not shown). As potency shows the average response size, when a release occurred, it adds new information. As the potency may be calculated from already obtained recordings, this additional analysis will strengthen the conclusions of the presented research

8.      It is well established that for main hippocampal synapses, the sign of long-term synaptic plasticity depends on frequency of applied stimulation. This dependency is characterized by BCM curve. In this light, presented shift of the LTD-LTP in knockout slices may stem from a change in crossover point of BCM curve or impairment of LTD. If the first scenario is correct, then even lower frequency stimulation than used (or shorter than 15 min.) may result in LTD. This direction of further research explaining the presented results is worth following. Authors should at least discuss it.

In my opinion editing of English language is not required

Reviewer 3 Report

In general, the study by Zheng et al. gives positive impression. However, there are several issues that have to be addressed before publication.

1.      The first problem that I see is that the major conclusions are drawn from the experiments with double knockouts of M1 and M3 receptors. Currently, the majority of researchers prefer to use conditional knockouts to avoid non-specific compensation that is developed during ontogeny in ordinary knockouts, which are used by the authors. The authors, however, use M1/M3 double knockout as animals that are the same in all aspects with wild type mice except for the expression of M1/M3 receptors in the hippocampus. It is not clear to which extent this statement is valid. Therefore, my propositions are as follows: (i) change the title of the article. “Long-Term but not Short-Term Plasticity at the Mossy Fiber - CA3 Pyramidal Cell Synapse in Hippocampus Is Altered in M1/M3 Double Knockout Mice” or something similar where you are not addressing to the conclusion that the authors would like to make based on their data.

(ii) Please, add section to the discussion on limitations of the study and discuss possible limitations derived from the model used in the study. It would also be logical to move a part of text in the results with references to Araya et al., 2006 about changes in M5-KO mice to this section. In my opinion, the authors should mention that pharmacological blockage of M1 and M3 receptors in wild-type animals or conditional knockout of these receptors in adult mice may give different result (or, instead, refer to papers which described similar results).

(iii)An alternative to (ii) is to make a series of experiments. If I understood correctly authors’ idea, acetylcholine is released from cholinergic fibers present in the slice during stimulation of mossy fibers by bipolar electrodes. In this case, optogenetic stimulation of dentate gyrus granule cells in slices from wild type animals (with dentate neurons transduced by AAV carrying channelrhodopsin) should give result similar to the result observed in M1/M3 knockout because under these conditions acetylcholine release will not be stimulated and, in the control animals, M1 and M3 receptors will not be activated. I believe that direct demonstration of the absence of M1/M3-mediated suppression of LTP would be a perfect support of authors’ conclusions. However, I leave this suggestion up to the authors.

2.      The description of statistical analysis is very short. It is not clear when and how ANOVA was used. For example, data presented in Figs. 2F, 4B, 4D, 5D, 5F should be compared by two-way ANOVA (factor 1 – time; factor 2 – knockout) and then, if possible, post hoc tests. The data of ANOVA should be presented in the text in the suitable form (with indication of degrees of freedom). Presumably, this analysis will not change the conclusions, however, the rest of the text and figures should be changed in accordance with the final results of this analysis.

Round 2

Reviewer 1 Report

I have no further request for the authors considering they addressed all my points

Author Response

We thank the reviewer for approving the Ms.

Reviewer 2 Report

Authors addressed all my concerns.

In my opinion the manuscript is ready for publication.

Author Response

(The authors gave the same response as above.)

Reviewer 3 Report

The only issue that remains in the manuscript is statistical analysis of data presented in Figures 2F, 4B, 5D and 5F. The authors mentioned in the reply that they used "the averaged magnitudes over 5 min (16-20 min) post induction protocol". This is possible way of data analysis, however, it is not correct. Again, they have to use two-way ANOVA for these datasets. Post hoc analysis (similar to what the authors used in manuscript) may be used only if statistical interaction between the factors is found. In their current analysis, they believe that this interaction is already present without any reason. Please, provide correct statistical analysis of your data. It is not very hard to do since practically any statistical software or online tools may be used to perform this analysis.

Author Response

We thank the reviewer for his/her comments. As requested, we have added the statistical analyses with two-way ANOVA for Figures 2F, 4B, 5D and 5F. Changes  in main text and legends are highlighted in red.